# Multiple health behaviour change interventions for primary prevention of cardiovascular disease in primary care: systematic review and meta-analysis

Samah Alageel, Martin C Gulliford, Lisa McDermott, Alison J Wright

Department of Primary Care and Public Health Sciences, King's College London, London, UK

**Correspondence to**
Samah Alageel; samah.alageel@kcl.ac.uk

## ABSTRACT

**Background** It is uncertain whether multiple behaviour change (MHBC) interventions are effective for the primary prevention of cardiovascular disease (CVD) in primary care. A systematic review and a meta-analysis were performed to evaluate the effectiveness of MHBC interventions on CVD risk and CVD risk factors; the study also evaluated associations of theoretical frameworks and intervention components with intervention effectiveness.

**Methods** The search included randomised controlled trials of MHBC interventions aimed at reducing CVD risk in primary prevention population up to 2017. Theoretical frameworks and intervention components were evaluated using standardised methods. Meta-analysis with stratification and meta-regression were used to evaluate intervention effects.

**Results** We identified 31 trials (36 484 participants) with a minimum duration of 12 months follow-up. Pooled net change in systolic blood pressure (16 trials) was −1.86 (95% CI −3.17 to −0.55; p=0.01) mm Hg; diastolic blood pressure (15 trials), −1.53 (−2.43 to −0.62; p=0.001) mm Hg; body mass index (14 trials), −0.13 (−0.26 to −0.01; p=0.04) kg/m²; serum total cholesterol (14 trials), −0.13 (−0.19 to −0.07; p<0.001) mmol/L. There was no significant association between interventions with a reported theoretical basis and improved intervention outcomes. No association was observed between intervention intensity (number of sessions and intervention duration) and intervention outcomes. There was significant heterogeneity for some risk factor analyses, leading to uncertain validity of some pooled net changes.

**Conclusions** MHBC interventions delivered to CVD-free participants in primary care did not appear to have quantitatively important effects on CVD risk factors. Better reporting of interventions' rationale, content and delivery is essential to understanding their effectiveness.

## Strengths and limitations of this study

▶ The review presents evidence of head-to-head meta-analysis of 31 published randomised controlled trials of multiple health behaviour change (MHBC) interventions and cardiovascular risk with follow-up for 12 months or longer.
▶ The study employed standardised instruments to evaluate the impact of theory use and behaviour change techniques in MHBC interventions.
▶ The majority of trials included were conducted in Europe and United States and only English language publications were included.
▶ Not all studies evaluated all outcomes of interest and some lacked detail concerning intervention design and delivery.

has informed the development of multiple health behaviour change (MHBC) interventions for reduction of CVD risk. Identifying individuals at high risk of CVD in primary care, and encouraging lifestyle change to reduce risk factors, represents a widely used strategy for the primary prevention of CVDs. Randomised controlled trials have been conducted in primary care to evaluate the effectiveness of MHBC interventions using lifestyle modification techniques instead of, or in addition to, pharmacological treatment to modify CVD risk factors. These trials have generally provided only equivocal evidence for reduction of CVD incidence through MHBC but the degree of effectiveness might be associated with level of risk.[5–7] Results from Ebrahim *et al.*'s[5] systematic review suggested that MHBC interventions have negligible effect on mortality in unselected populations, with a pooled OR for coronary heart disease mortality of 0.99 (95% CI 0.92 to 1.07). Evidence of benefit was found in studies in high-risk populations including people with hypertension (OR 0.78; 0.68 to 0.89) or diabetes (OR 0.71; 0.61 to 0.83).[5]

## INTRODUCTION

Cardiovascular disease (CVD) is the leading cause of death worldwide, accounting for over 30% of global mortality.[1] CVD is mediated by several antecedent behavioural risk factors, and its onset might be prevented or delayed by altering one or several risk factors.[1] Risk factors for CVD are inter-related and often coexist.[2–4] This observation

However, general health checks were not found to reduce all-cause mortality, nor CVD-related or cancer-related morbidity and mortality.[8]

Previous reviews have assessed the effectiveness of MHBC interventions in reducing CVD morbidity and mortality,[5 6 8] and less is known about the effectiveness of these interventions in reducing CVD risk and risk factor values in primary care.

In recent years, there has been growing appreciation of the role of employing psychological theory in behaviour change intervention design and studying the impact of specific behaviour change techniques (BCTs) on intervention outcomes.[9] Theories of the psychological determinants of behaviour can be used to inform the development and evaluation of behaviour change interventions.[10] Interventions are likely to be more effective when they systematically target psychological determinants of behaviour.[11] A review of internet-based interventions suggested that more intensive use of theory was associated with greater behaviour change,[12] but another review found little evidence of an association between theory use and intervention effects on healthy eating or physical activity.[13] This equivocal evidence could arise if a high proportion of behaviour change interventions are not based on a theory or the theory is not applied extensively.[14]

BCTs are 'the active components of an intervention designed to change behaviour'.[15] Identifying specific BCTs associated with greater impact on intervention effectiveness is essential for future intervention design.[16] Previous reviews suggested that interventions using the BCTs 'provision of instructions,' 'self-monitoring of behaviour,' 'relapse prevention' and 'prompt practice' led to greater reductions in weight among obese individuals,[17] while interventions designed to modify physical activity and/or diet were more effective when they included self-monitoring plus one of the four following BCTs: prompting intention formation, specific goal setting, review of behavioural goals or providing feedback on performance.[18] Identifying BCTs associated with greater intervention effectiveness and exploring the impact of applying theory will contribute to the design of future MHBC interventions targeting CVD risk in primary care.

## Objectives

This systematic review had three objectives: first, to assess the effectiveness of MHBC interventions, directed at changing two or more behaviours, at reducing CVD risk and CVD risk factors in adults without existing cardiovascular conditions; second, to evaluate whether using theory to develop interventions is associated with intervention effectiveness; and third, to evaluate the association between BCTs employed and intervention effects.

## METHODS

Studies were selected according to the criteria mentioned in the below sections.

## Participants

Trials that recruited an adult population (>18 years old) free of CVD were included. Following previous reviews,[5] we included trials with less than 20% participants with CVD. Studies of patient populations with established disease, such as diabetes, were excluded.

## Interventions

We included studies that evaluated behaviour change interventions aimed at reducing CVD risk by intervening on two or more risk behaviours at the same time. Risk behaviours included the following: physical activity, diet, alcohol consumption, use of stress management and smoking. Comparators were usual care or less intensive interventions.

## Settings

Interventions where participants were recruited, and interventions were delivered by trained healthcare professionals or primary care staff, in primary care premises (including general practice, family practice or primary care clinic).

## Study design

Controlled trials, with individual or cluster randomisation, providing ≥12 months follow-up were used for outcome evaluation.

## Outcome measures

Long-term outcomes of MHBC interventions including CVD mortality and clinical events have been reported previously[5 6] and only one study in 2015 included clinical events as an outcome. Therefore, long-term outcomes were not included in this systematic review. Primary outcomes were changes in CVD risk scores, body mass index (BMI) or body weight, blood pressure and serum total cholesterol levels. We have excluded diabetes management trials; therefore, diabetes control outcomes were not included. Secondary outcomes were changes in physical activity, diet, smoking and alcohol consumption.

## Language

Studies reported in English.

## Search strategy

Multiple sources of ascertainment were used, including electronic databases (Medline, EMBASE, PsycINFO and CENTRAL) and searching reference lists of included papers. The search results and search terms of previous review[5] were used with searching extended from 2006 until February 2017. Search terms used included primary prevention, multiple risk factor, lifestyle intervention, health education and health promotion. (Online supplementary appendix A presents the search strategies used.) Titles were screened by one reviewer (SA) and a second reviewer (MG) checked a random set of studies, approximately 10% of the search results, to assess agreement regarding whether they met the inclusion criteria. Disagreements were resolved through discussion, until

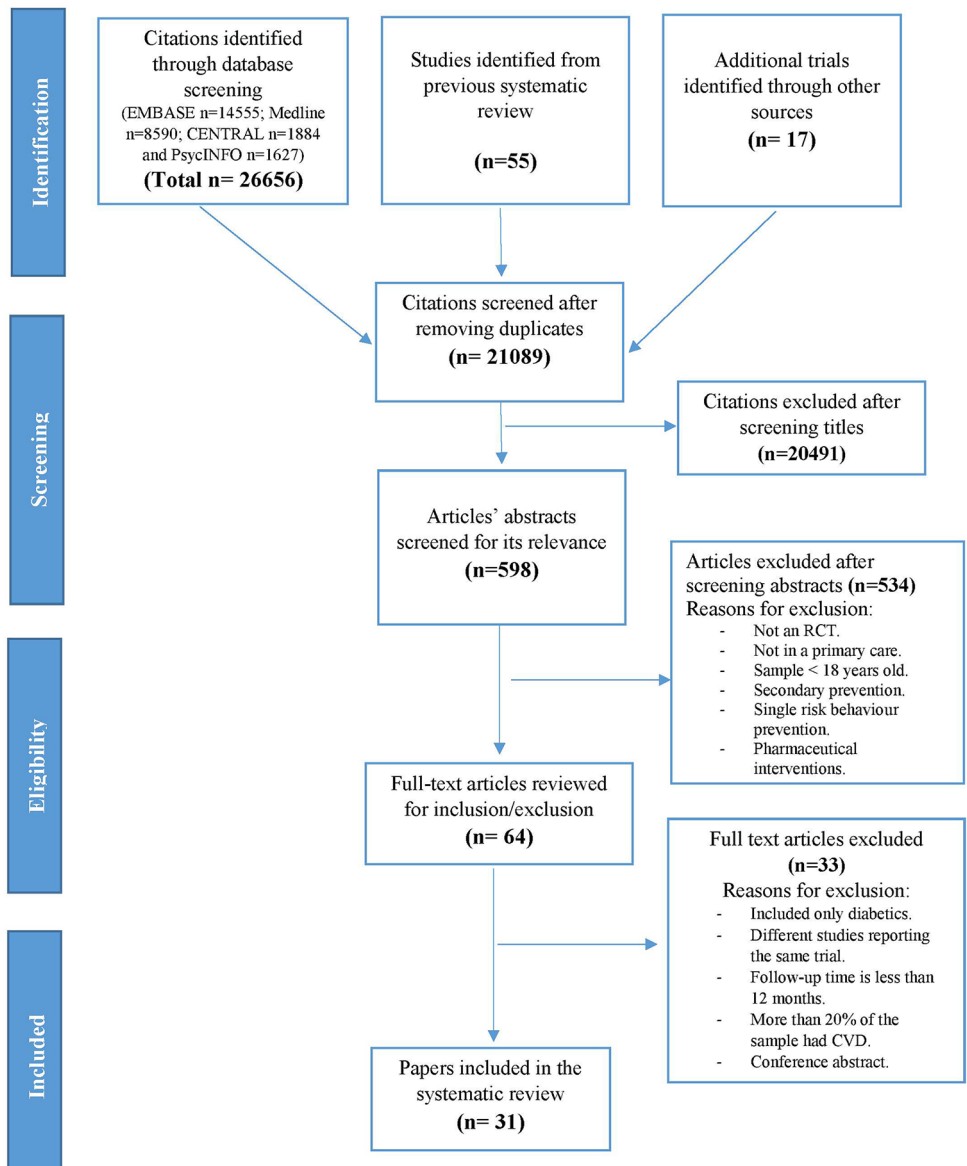

**Figure 1** Preferred Reporting Items for Systematic Reviews and Meta-Analyses flow diagram outlining the systematic review process.

full agreement was reached. The selection process is displayed in figure 1.

## Methodological quality

Studies were evaluated using the Cochrane risk of bias tool.[19] This assesses six domains of bias including selection bias, performance bias, detection bias, attrition bias, reporting bias and other biases.[19]

## Data extraction

Interventions were coded by country, target behaviours, participant and intervention characteristics, mode of delivery and intervention outcomes. We attempted to contact study authors to provide additional information where necessary. However, when information was not available, we assumed missing outcome data to occur at random.

In addition, Michie and Prestwich's[20] method of assessing the application of theory in the development and evaluation of behaviour change interventions was used. The Theory Coding Scheme (TCS) consists of 19 items that cover different aspects that may be informed by theory.[20] We used three measures to capture the extent of theory use, as employed in a previous review[13]: the first concerned whether the intervention was explicitly based on a theory or combination of theories or predictors (TCS item 5). Second, we assessed the degree to which each BCT reported as part of the intervention was linked to a theory-relevant construct (scored +2 for the ideal scenario of 'yes' to TCS item 7 (all intervention techniques explicitly linked to at least one theory-relevant construct), +1 for studies coded 'yes' for TCS item 8 (at least one, but not all, intervention technique explicitly

linked to at least one theory-relevant construct) and/ or TCS item 9 (group of BCTs are linked to a group of constructs) and 0 for studies coded 'no' for all of items 7–9). Finally, we rated the extent to which all constructs in the relevant theory had been explicitly targeted by BCTs. This was scored +2 for the ideal scenario of 'yes' to TCS item 10 (all theory-relevant constructs explicitly linked to at least one BCT), +1 for 'yes' to TCS item 9 (group of BCTs are linked to a group of constructs) and/or item 11 (at least one, but not all, theory-relevant construct is explicitly linked to at least one BCT) and 0 for interventions coded 'no' to all of items 9–11.

The theory-based taxonomy of 93 BCTs developed by Michie et al[9] was used to identify intervention techniques. The assessment was completed by two researchers (LM and SA) with good agreement for intervention groups (77.8% agreement) and control groups (92.6% agreement). Discrepancies were discussed and resolved to reach full agreement. Intervention characteristics and BCTs were also extracted from descriptions of the control group because the chosen nature of the control group can influence the apparent effectiveness of interventions.[21] Where detail of interventions was lacking, we attempted to contact study authors to provide additional information.

### Data analysis

Outcome data were combined in random effects meta-analyses using 'metan' commands in STATA. DerSimonian and Laird[22] random effect models were chosen due to the considerable heterogeneity for certain outcomes. For continuous outcomes, we used mean changes in each trial arm to calculate net effects. We expressed effects for binary variables as risk differences. We quantified statistical heterogeneity using $I^2$ statistic. We have examined the influence of individual studies in outcomes with considerable heterogeneity ($I^2 > 50\%$) by omitting one study at a time to see the extent to which heterogeneity could be explained by a study or group on studies (leave-one-out analysis).

Meta-regression analyses were used to examine the effect of medication use, number of interventions' sessions, intervention duration, types of BCTs used and theory use on intervention outcomes. Intervention duration was calculated by multiplying the number of sessions and the sessions' duration. Publication bias was assessed using Egger's regression test[23] using 'metabias' and 'metafunnel' commands in STATA. If bias existed, the 'trim and fill'[24] method was used to adjust for publication bias. Mendis et al[25] Nigeria site's study had unusually high summary estimates, and heterogeneity diminished substantially after this study was excluded. This study was therefore treated as an outlier and results were reported with the exclusion of this study.

### RESULTS

The initial search identified 26656 references, with 55 relevant trials identified from the previous systematic review.[5] After removing duplicates, 21089 titles were

**Table 1** Summary of characteristics of 27 trials included in the review

| Characteristics | | Freq. (%) |
|---|---|---|
| **Total** | | **31 (100)** |
| Country | UK | 6 (19.4%) |
| | Sweden | 5 (16.1%) |
| | Netherlands | 4 (12.9%) |
| | USA | 4 (12.9%) |
| | Europe | 7 (22.6%) |
| | Others | 5 (16.1%) |
| Participants (n) | Median (IQR) | 419 (224–883) |
| Gender | Male only | 1 (3.2) |
| | Female only | 1 (3.2) |
| | Both | 29 (93.5) |
| Age | Minimum age, median (IQR) | 30 (20–40) |
| | Maximum age, median (IQR) | 65 (60–74) |
| Intervention outcomes | CVD risk | 14 (45.2) |
| | Body weight | 25 (80.6) |
| | Blood pressure | 26 (83.9) |
| | Serum cholesterol | 26 (83.9) |
| | Diet | 18 (58.1) |
| | Physical activity | 21 (67.7) |
| | Alcohol | 6 (19.4) |
| | Smoking | 15 (48.4) |
| Targeted behaviours (n) | Two behaviours | 11 (35.5) |
| | Three behaviours | 12 (38.7) |
| | Four behaviours | 7 (22.6) |
| | Five behaviours | 1 (3.2) |
| Follow-up duration | 12 months | 18 (58.1) |
| | >12 months | 13 (41.9) |

Figures are frequencies (column percent).
CVD, cardiovascular disease.

screened. A total of 31 trials were included in this review (figure 1).

### Included studies

We identified a total of 31 trials of MHBC intervention for the primary prevention of CVD in primary care with 36 484 participants. The duration of follow-up ranged from 12 months to 6 years (median=12 months). Intervention duration ranged from 2 months up to 3 years (median=12 months). Summary of included studies characteristics is presented in table 1 and online supplementary table 1.

### Study characteristics

Diet and physical activity were targeted in 11 trials, with nine trials targeting diet, physical activity and smoking. Diet, physical activity, smoking and alcohol consumption were targeted in seven interventions and two

**Table 2** Summary of interventions characteristics for 27 trials included in the review

| | | Intervention n (%) | Control n (%) |
|---|---|---|---|
| Type of staff delivering intervention | General practitioners and physicians | 10 (32.3) | |
| | Nurses | 15 (48.4) | |
| | Dietitian | 7 (22.6) | |
| | Others | 12 (38.7) | |
| Mode of intervention delivery | Face-to-face sessions | 30 (96.8) | 14 (45.2) |
| | Group sessions | 9 (29.0) | 1 (3.2) |
| | Written materials | 15 (48.4) | 7 (22.6) |
| | Telephone sessions | 8 (25.8) | – |
| | Unclear | – | 13 (41.9) |
| Intervention sessions (n) | 1–4 sessions | 5 (16.1) | 9 (29.0) |
| | 5–9 sessions | 11 (35.5) | 2 (6.5) |
| | 10–15 sessions | 4 (12.9) | 1 (3.2) |
| | >15 sessions | 5 (16.1) | 1 (3.2) |
| | Unclear | 6 (19.4) | 18 (58.1) |
| BCTs (n) | 1–2 BCTs | 5 (16.1) | 14 (45.2) |
| | 3–4 BCTs | 10 (32.3) | 1 (3.2) |
| | 5–6 BCTs | 12 (38.7) | – |
| | 7–9 BCTs | 3 (9.7) | – |
| | 10 BCTs | 1 (3.2) | – |
| | Unclear | – | 16 (51.6) |
| Frequently used BCTs | Credible source (9.1) | 22 (70.9) | 6 (19.4) |
| | Goal setting (behaviour) (1.1) | 19 (61.3) | 2 (6.5) |
| | Information about health consequences (5.1) | 9 (29.0) | 5 (16.1) |
| | Instruction on how to perform a behaviour (4.1) | 9 (29.0) | 1 (3.2) |
| | Action planning (1.4) | 9 (29.0) | – |
| | Self-monitoring of behaviour (2.3) | 8 (25.8) | – |

Figures are frequencies (column percent)
BCT, behaviour change technique.

interventions targeted diet, physical activity and stress management. Only one intervention targeted diet, physical activity, stress and alcohol consumption and one intervention targeted all five behaviours. A wide range of intervention modalities was investigated (table 2 and online supplementary table 2), including individual and group sessions, telephone conversations and provision of written materials. The majority of the included trials reported offering 'usual care' to the control group, with few details provided. Seven trials offered face-to-face sessions and seven trials offered face-to-face sessions and written materials. Written materials alone were offered in three trials and no intervention was offered to the control group in three interventions.

### Risk of bias in included studies

Risk of bias assessment is presented (online supplementary table 3). Half of the included trials (n=16) reported using

intention-to-treat (ITT) analysis, while 15 studies did not state ITT procedures. Loss to follow-up ranged from 1.5% to 50.9%. Random allocation methods were not usually reported. In only 14 out of 31 trials, the method used was considered adequate. It is not possible to blind participants and personnel to treatment allocation in lifestyle intervention, which raises the possibility of bias inevitably. Only five trials have reported blinding of participants and personnel. Eleven trials have reported blinding outcomes assessors to treatment allocation; this too makes the assessment of outcomes likely biased (eg, self-reported outcomes). Not all trials reported sufficient detail to assess risk of bias and these were rated as 'unclear'.

### Treatment fidelity

Few studies reported using fidelity checks[26–30] to confirm that interventions were delivered as intended and this

raises a question of whether the interventions were delivered as planned and in a consistent manner.

## Effect of interventions

Pooled effect sizes for all outcomes are presented in table 3 and forest plots are presented (online supplementary appendix B).

### Changes in CVD risk factors

#### Blood pressure

Sixteen trials[25 27 31–44] reported changes in participants' systolic blood pressure (SBP) with no evidence of publication bias (Egger's test, p=0.79). The weighted mean difference in SBP was −1.86 mm Hg (95% CI −3.17 to −0.55 mm Hg; p=0.01). Diastolic blood pressure (DBP) was reported in 15 trials,[25 27 31–33 35–44] with no evidence of publication bias (Egger's test, p=0.19). Weighted mean difference in DBP was −1.53 mm Hg (−2.43 to −0.62 mm Hg; p=0.001). Out of the 12 interventions that evaluated blood pressure, seven reported that participants in all study groups were taking antihypertensive medications and three reported that they were taking unspecified medications. There are no significant differences between the impact of trials that reported use of medication on SBP (β=−1.72; p=0.23) and DBP (β=−1.46; p=0.12) compared with trials that did not report using medications.

#### Serum total cholesterol

Fourteen trials[27 31–33 35–37 39–45] evaluated serum total cholesterol and provided sufficient data for analysis (Egger's test, p=0.55). Serum total cholesterol levels showed a small decrease in favour of intervention (−0.13 mmol/L; 95% −0.19 to −0.07; p<0.001). Six of the trials included in the analysis reported the use of lipid lowering medication and two reported the use of unspecified medication by all study groups. The weighted mean difference for total cholesterol was not different between trials that reported using medication and trials that have not stated using medications (β=0.01; p=0.75) (table 3).

#### Smoking

Eleven studies[25 26 29–32 34 37 44 46 47] reported smoking prevalence following the intervention. The pooled analysis showed no evidence of reductions in smoking behaviour (risk difference −0.00%; 95% CI −0.02 to 0.01; p=0.66). All studies included in the analysis relied on self-reported smoking status and only two[29 44] reported using smoking cessation medication. There was no evidence of publication bias (p=0.47).

#### Weight and BMI

Fourteen studies[25 27 31 33 35–44] reported on BMI as an outcome. The weighted mean change was −0.13 kg/m$^2$ (95% CI −0.26 to −0.01; p=0.04). The results of 'trim and fill' method indicated that the weighted mean did not change despite the existence of publication bias (Egger's test, p=0.002). Fewer studies (n=12)[27 33 35–37 40–44 47 48] reported on weight changes, showing a reduction of −0.91 kg (CI

−1.39 to −0.43 kg; p<0.001) with no evidence of publication bias (p=0.97).

#### Dietary behaviour

Sixteen trials[25–30 33 34 42–45 47–50] reported dietary behaviours as an outcome of the interventions. Outcomes of dietary interventions were measured using diverse methods; therefore, a meta-analysis was not conducted. Trials used a range of dietary self-report instruments to assess dietary behaviour, and none has used additional objective measures. Fruit and vegetable consumption was reported either as portions per day[25–27 43 47 50] or as proportion of participants who met the recommendation for fruits and vegetable intake.[26 33 34] There was no positive effect of the intervention on fruits and vegetable consumption in most of the trials,[26 27 34 47] and some trials did report improvement following the intervention.[33 43 50] Fat intake was commonly measured as a dietary outcome either in terms of fat intake per day[27 33 48 49] or as a fat score.[29 34] All the trials reported reductions in fat intake after the intervention, except Koelewijn-van Loon et al[34] trial, where there was no significant difference between the intervention and control group.

#### Physical activity behaviour

Twenty trials reported changes in physical activity.[26–34 36 37 39 40 42–44 46 47 50 51] Physical activity was assessed via self-report. Due to the variety of measurements used, meta-analysis was not feasible. Some trials reported physical activity as the proportion of participants who are physically active.[29 31 39 46 50] Other studies measured physical activity as the number of minutes per week,[27 34 42 43] or classified participants based on their weekly exercise.[26 28 44 50] Eight of these trials[27 29 30 36 37 40 42 44 47 50] resulted in an increase in reported physical activity following the intervention, and nine[26 28 31–33 39 43 46 48 51] trials concluded that the intervention had no impact on physical activity.

#### Alcohol consumption

Alcohol consumption was reported as an outcome in seven trials.[30 34 40 46–48 50] However, it was measured differently, which did not allow for pooled effect analysis. Two trials[40 46] reported reductions in alcohol consumption following the interventions, whereas the majority of the studies[30 34 47 48 50] did not find significant reductions in alcohol intake.

#### CVD risk

Studies used different risk scores to examine the effect of interventions on CVD risk. Two studies[38 51] used the Framingham risk equation,[52] two studies[30 53] used the Dundee Risk Score[54] and one study[44] used QRISK2 score.[55] These trials reported larger CVD risk reductions in the intervention group compared with the control group. All of these trials had missing data making it not possible to analyse the pooled effect. Four studies[26 34 39 46] used the SCORE risk;[56] however, because of missing data, we only included two studies[26 34] in the analysis, both conducted in the Netherlands. There was a non-significant increase

**Table 3** Pooled effects from meta-analysis of multiple health behaviour interventions on CVD risk and CVD risk factors

| Outcome | n | | Pooled effect size | 95% CI | p Value | I² (%) | p Value for heterogeneity | τ² |
|---|---|---|---|---|---|---|---|---|
| Systolic blood pressure (mm Hg) | 16 | | −1.86 | −3.17 to −0.55 | 0.01 | 63.0 | <0.001 | 3.91 |
| Systolic blood pressure (mm Hg) by medication use | 10 | Medication | −2.59 | −4.48 to −0.69 | 0.01 | 68.3 | 0.001 | 5.31 |
| | 6 | None* | −0.55 | −1.69 to 0.59 | 0.35 | 3.4 | 0.40 | 0.09 |
| Diastolic blood pressure (mm Hg) | 15 | | −1.53 | −2.43 to −0.62 | 0.001 | 68.3 | <0.001 | 1.92 |
| Diastolic blood pressure (mm Hg) by medication use | 10 | Medication* | −1.96 | −2.79 to −1.11 | <0.001 | 42.5 | 0.07 | 0.66 |
| | 5 | None | −0.78 | −2.50 to 0.93 | 0.37 | 73.0 | <0.001 | 2.97 |
| Serum total cholesterol (mmol/L) | 14 | | −0.13 | −0.19 to −0.07 | <0.001 | 20.3 | 0.22 | 0.0 |
| Serum total cholesterol (mmol/L) by medication use | 8 | Medication | −0.15 | −0.26 to −0.03 | 0.01 | 43.8 | 0.09 | 0.01 |
| | 6 | None* | −0.11 | −0.18 to −0.03 | 0.01 | 0.0 | 0.60 | 0.0 |
| Smoking (%) | 11 | | −0.00 | −0.02 to 0.01 | 0.66 | 13.4 | 0.31 | 0.0 |
| BMI (kg/m²) | 14 | | −0.13 | −0.26 to −0.01 | 0.04 | 0.0 | 0.82 | 0.0 |
| Body weight (kg) | 10 | | −0.91 | −1.39 to −0.43 | <0.001 | 12.1 | 0.33 | 0.08 |
| CVD risk using SCORE (%) | 2 | | 0.12 | −0.37 to 0.61 | 0.61 | 0.0 | 0.87 | 0.0 |
| Systolic blood pressure (mm Hg) by theory use | 5 | Theory | −2.18 | −5.92 to 1.56 | 0.25 | 72.3 | 0.01 | 13.0 |
| | 11 | None† | −1.69 | −3.01 to −0.29 | 0.02 | 61.3 | <0.01 | 3.01 |
| Diastolic blood pressure (mm Hg) by theory use | 5 | Theory | −1.25 | −2.43 to −0.06 | 0.04 | 0.4 | 0.40 | 0.01 |
| | 10 | None† | −1.67 | −2.83 to −0.52 | <0.001 | 76.9 | <0.001 | 2.42 |
| Serum total cholesterol (mmol/L) by theory use | 4 | Theory | −0.03 | −0.15 to 0.10 | 0.68 | 0.0 | 0.48 | 0.0 |
| | 10 | None† | −0.13 | −0.20 to −0.07 | <0.001 | 0.0 | 0.29 | 0.0 |
| BMI (kg/m²) by theory use | 5 | Theory | −0.15 | −0.41 to 0.10 | 0.24 | 0.0 | 0.96 | 0.0 |
| | 9 | None† | −0.13 | −0.28 to 0.02 | 0.10 | 0.0 | 0.44 | 0.0 |
| Body weight by (kg) theory use | 4 | Theory | −0.24 | −0.94 to 0.45 | 0.49 | 0.0 | 0.97 | 0.0 |
| | 8 | None† | −1.32 | −1.80 to −0.83 | <0.001 | 0.0 | 0.53 | 0.0 |

*Medication use is not reported.
†Theory use is not reported.
BMI, body mass index; CVD, cardiovascular disease; I², index of heterogeneity; N, number of trials; SCORE, Systematic COronary Risk Evaluation.

in weighted mean difference of 0.12% CVD risk (95% CI −0.37 to 0.61; p=0.62).

## Sensitivity analysis

In outcomes of considerable heterogeneity ($I^2$ >50%), we sought to identify possible causes by exploring the effect of included studies using leave-one-out sensitivity analysis. The absence of study Mendis et al (China site)[25] and Koelewijn-van Loon et al[34] in analysing the impact of interventions of SBP reduces heterogeneity from $I^2$=63% to $I^2$=49.4% and generated a weighted mean difference (−1.86; CI −3.17 to −0.54; p=0.001) similar to the one obtained with all 16 trials. For DBP, removing Knutsen and Knutsen[31] from the analysis has resulted in reducing heterogeneity from $I^2$=68.3% to $I^2$=37.8% and produced a larger weighted mean difference (−1.93; CI −2.69 to −1.18; p<0.001).

## Intervention components

### Intervention time and number of sessions

The number of sessions was reported in 24 trials, ranging from 3 to 56 sessions (median=6 sessions). No significant associations were detected between the number of sessions and SBP ($\beta$=−0.17; p=0.15), DBP ($\beta$=−0.15; p=0.08), BMI ($\beta$=−0.01; p=0.57) and weight ($\beta$=0.02; p=0.68). Interventions with more sessions were associated with slight reductions in serum total cholesterol ($\beta$=−0.01; p=0.02). Thirteen of the included trials provided enough details to calculate intervention delivery duration, which ranged from 45 min to 2.5 hours (median=300 min). No significant associations were detected between intervention duration and SBP ($\beta$=−0.00; p=0.26), DBP ($\beta$=−0.00; p=0.45), BMI ($\beta$=−0.00; p=0.53) and weight ($\beta$=−0.00; p=0.55). Hence, more sessions and longer intervention duration were not necessarily associated with greater intervention effectiveness.

### Theory use

Of the 31 trials included, nine reported some use of psychological theory (or a combination of two theories) in relation to the intervention. The Transtheoretical Model[57] was used in eight trials,[27 28 36–40 47] while Social Cognitive Theory[58] was used in four[27 28 38 59] interventions.

We tested the extent of theory use using TCS[20] in three ways (see online supplementary table 4). The first method was based on the use of theory in selecting intervention techniques (item 5 in TCS). Only four trials were coded yes for this item. The second method was used to reflect the extent to which reported BCTs were linked to theory-relevant constructs (items 7–9). Only four trials were coded yes to at least one of these items. The third method was used to reflect the extent to which all theory-relevant constructs were targeted by BCTs (items 9–11). Only four trials were coded yes to at least one of these items. Therefore, we were not able to examine the impact of differing levels of theory use on intervention outcomes due to the small number of trials using theory extensively. However, we were able to test whether studies that merely reported using a theory had greater impact on outcomes using meta-regression. There was no significant association between studies which reported using a theory and SBP ($\beta$=−0.13; p=0.89), DBP ($\beta$=−0.37; p=0.73) and BMI ($\beta$=−0.03; p=0.87). Studies that reported using a theory had increased weight ($\beta$=1.07; p=0.03; CI 0.11 to 2.04) and serum total cholesterol outcomes ($\beta$=0.19; p=0.04) compared with studies that did not report using a theory.

### Effectiveness of specific BCTs

The number of BCTs in the intervention group varied, ranging from 2 to 10 BCTs (median=5). BCTs in the control group were generally poorly described as the majority of trials (n=16) did not appear to offer any BCTs.

Twenty-nine different BCTs were identified from the included trials (see online supplementary table 2). The most commonly used BCTs in the intervention group were 'credible source' and 'Goal setting (behaviour)', which were used in 22 and 19 trials, respectively. In the control group, 'Credible source' and 'Information about health consequences' were most commonly used, which were used in six and five interventions, respectively.

We tested the potential impact of using specific BCTs on intervention outcomes (table 4). For SBP, one BCT had a significant influence on effect sizes. Interventions employing 'Review of behaviour goal(s)' resulted in an increase in SBP ($\beta$=3.45; p=0.04) compared with those not using this BCT. For DBP and total cholesterol, there were no BCTs significantly associated with the effectiveness of the interventions. The same was the case for BMI, but for weight, interventions that included 'Action planning' resulted in greater reductions than those that did not ($\beta$=−1.10; p=0.04).

## DISCUSSION

This systematic review is among the first to evaluate the impact of theory use and BCTs in MHBC interventions for reducing CVD risk, although pooled effects of interventions on risk factors were statistically significant but clinically modest. The results of this systematic review suggest that MHBC interventions evaluated to date for the primary prevention of CVD may generally have very limited effects in reducing CVD risk and CVD risk factors in primary care populations.

Previous systematic reviews have investigated the effectiveness of interventions aimed at individual risk factors including diet, physical activity and body weight.[6 60] These reviews generally find that behaviour change interventions in primary care have minor impact on risk factor values. The Cochrane review up to 2011 reported modest reductions in CVD risk factors following MHBC interventions that were slightly greater than we report.[5] However, the Cochrane review did not restrict the intervention setting to primary care.

Estimated changes in CVD risk factors should be viewed with caution. In the present set of trials, the average duration of follow-up was 12 months and changes in risk

**Table 4** Meta-regression results of intervention effects for studies using or not using particular BCTs.

| Outcome | BCT | BCT included | | | BCT not included | | | | | |
|---|---|---|---|---|---|---|---|---|---|---|
| | | MD | CI | n | MD | CI | n | β | CI | p Value |
| Systolic blood pressure | 1.1 Goal setting (behaviour). | −1.12 | −2.49 to 0.25 | 11 | −3.87 | −5.07 to −2.67 | 5 | 2.79 | −0.19 to 5.78 | 0.07 |
| | 1.2 Problem solving. | −3.19 | −9.21 to 2.83 | 3 | −1.69 | −2.99 to −0.39 | 13 | −0.98 | −5.12 to 3.16 | 0.62 |
| | 1.3 Goal setting (outcome). | −3.01 | −6.99 to 0.97 | 3 | −1.65 | −3.12 to −0.18 | 13 | −1.11 | −4.95 to 2.73 | 0.55 |
| | 1.4 Action planning. | −2.84 | −5.29 to −0.39 | 7 | −1.39 | −3.01 to 0.23 | 9 | −1.39 | −4.54 to 1.76 | 0.36 |
| | **1.5 Review behaviour goal(s)** | **0.93** | **−2.10 to 3.95** | **4** | **−2.49** | **−3.82 to −1.16** | **12** | **3.45** | **0.13 to 6.76** | **0.04** |
| | 4.1 Instruction on how to perform the behaviour. | −2.77 | −4.89 to −0.68 | 5 | −1.23 | −2.89 to 0.44 | 11 | −1.53 | −4.53 to 1.47 | 0.29 |
| | 5.1 Information about health consequences. | −0.70 | −2.13 to 0.73 | 5 | −2.59 | −4.43 to −0.76 | 11 | 1.75 | −1.15 to 4.65 | 0.22 |
| | 9.1 Credible source. | −2.75 | −4.34 to −1.17 | 9 | −0.43 | −2.81 to 1.96 | 7 | −2.47 | −5.51 to 0.58 | 0.10 |
| | 9.2 Pros and cons. | 0.16 | −3.89 to 4.20 | 4 | −2.31 | −3.59 to −1.02 | 12 | 2.67 | −0.84 to 6.18 | 0.13 |
| | 11.2 Reduce negative emotions. | −3.52 | −4.93 to −2.11 | 4 | −1.05 | −2.46 to 0.37 | 12 | −0.22 | −5.48 to 5.03 | 0.93 |
| Diastolic blood pressure | 1.1 Goal setting (behaviour). | −1.18 | −2.31 to −0.04 | 10 | −3.37 | −3.78 to −0.96 | 5 | 1.35 | −0.89 to 3.60 | 0.22 |
| | 1.4 Action planning. | −2.17 | −4.13 to −0.20 | 6 | −1.28 | −2.30 to −0.25 | 9 | −0.82 | −3.02 to 1.38 | 0.44 |
| | 2.3 Self-monitoring of behaviour. | −1.29 | −3.03 to 0.46 | 3 | −1.58 | −2.60 to −0.55 | 12 | 0.18 | −2.73 to 3.08 | 0.89 |
| | 4.1 Instruction on how to perform the behaviour. | −1.12 | −2.80 to 0.56 | 5 | −1.84 | −2.77 to −0.91 | 10 | 0.87 | −1.17 to 2.91 | 0.38 |
| | 5.1 Information about health consequences. | −0.64 | −2.52 to 1.24 | 4 | −1.92 | −2.79 to −1.05 | 11 | 1.46 | −0.53 to 3.46 | 0.14 |
| | 9.1 Credible source. | −1.85 | −3.44 to −0.26 | 6 | −1.29 | −1.91 to −0.68 | 6 | −0.37 | −2.51 to 1.77 | 0.72 |
| | 9.2 Pros and cons. | −1.46 | −3.05 to 0.13 | 3 | −0.93 | −1.46 to 0.12 | 12 | 0.04 | −2.73 to 2.80 | 0.98 |
| | 11.2 Reduced negative emotions. | −2.98 | −4.71 to −1.25 | 4 | −1.25 | −2.25 to −0.25 | 11 | −1.78 | −4.46 to 0.89 | 0.17 |

**Table 4** Continued

| Outcome | BCT | BCT included | | | BCT not included | | | | | | |
| | | MD | CI | n | MD | CI | n | β | CI | p Value |
|---|---|---|---|---|---|---|---|---|---|---|
| Serum total cholesterol | 1.1 Goal setting (behaviour). | −0.11 | −0.17 to −0.5 | 9 | −0.17 | −0.34 to −0.01 | 5 | 0.09 | −0.08 to 0.26 | 0.29 |
| | 1.3 Goal setting (outcome). | −0.21 | −0.45 to 0.02 | 3 | −0.10 | −0.16 to −0.03 | 11 | −0.12 | −0.27 to 0.03 | 0.11 |
| | 1.4 Action planning. | −0.15 | −0.34 to 0.05 | 6 | −0.12 | −0.18 to −0.06 | 8 | −0.05 | −0.22 to −0.12 | 0.52 |
| | 4.1 Instruction on how to perform the behaviour. | −0.18 | −0.19 to −0.07 | 4 | −0.10 | −0.17 to −0.04 | 10 | −0.09 | −0.25 to 0.06 | 0.20 |
| | 5.1 Information about health consequences. | −0.11 | −0.18 to −0.07 | 5 | −0.15 | −0.28 to −0.03 | 9 | 0.07 | −0.08 to −0.22 | 0.35 |
| | 9.1 Credible source. | −0.12 | −0.21 to −0.04 | 9 | −0.14 | −0.22 to −0.05 | 5 | −0.00 | −0.18 to 0.17 | 0.96 |
| | 9.2 Pros and cons. | −0.07 | −0.31 to 0.16 | 3 | −0.14 | −0.20 to −0.07 | 11 | 0.05 | −0.18 to 0.28 | 0.64 |
| BMI | 1.1 Goal setting (behaviour). | −0.24 | −0.42 to −0.05 | 10 | −0.03 | −0.21 to 0.15 | 4 | −0.20 | −0.49 to 0.08 | 0.14 |
| | 1.3 Goal setting (outcome). | −0.09 | −0.50 to 0.32 | 3 | −0.14 | −0.27 to −0.00 | 11 | 0.05 | −0.42 to 0.52 | 0.83 |
| | 1.4 Action planning. | −0.32 | −0.61 to −0.04 | 6 | −0.09 | −0.23 to 0.06 | 8 | −0.24 | −0.59 to 0.11 | 0.17 |
| | 1.5 Review behaviour goal(s) | −0.66 | −1.51 to 0.20 | 3 | −0.12 | −0.25 to 0.01 | 11 | −0.54 | −1.48 to 0.41 | 0.25 |
| | 4.1 Instructions on how to perform the behaviour. | −0.07 | −0.23 to 0.09 | 5 | −0.24 | −0.45 to −0.03 | 9 | 0.17 | −0.12 to 0.46 | 0.23 |
| | 5.1 Information about health consequences. | −0.13 | −0.36 to 0.10 | 4 | −0.14 | −0.29 to 0.02 | 10 | 0.01 | −0.29 to 0.31 | 0.96 |
| | 9.1 Credible source. | −0.24 | −0.43 to −0.06 | 8 | −0.03 | −0.21 to 0.07 | 6 | −0.21 | −0.49 to 0.07 | 0.13 |
| | 9.2 Pros and cons. | −0.47 | −1.29 to 0.34 | 3 | −0.13 | −0.26 to 0.01 | 11 | −0.35 | −1.25 to 0.55 | 0.42 |
| | 11.2 Reduce negative emotions. | −0.33 | −0.79 to 0.14 | 3 | −0.12 | −0.25 to 0.02 | 11 | −0.21 | −0.74 to 0.32 | 0.41 |

**Table 4** Continued

| Outcome | BCT | BCT included | | | BCT not included | | | | | |
|---|---|---|---|---|---|---|---|---|---|---|
| | | MD | CI | n | MD | CI | n | β | CI | p Value |
| Weight | 1.3 Goal setting (outcome). | −1.02 | −1.73 to −0.31 | 4 | −0.83 | −1.53 to −0.12 | 8 | −0.17 | −1.33 to 0.99 | 0.75 |
| | **1.4 Action planning.** | **−1.27** | **−1.74 to −0.79** | **7** | **−0.17** | **−0.94 to 0.58** | **5** | **−1.10** | **−2.11 to −0.09** | **0.04** |
| | 1.5 Review behaviour goal(s). | −1.67 | −4.77 to 1.40 | 3 | −0.86 | −1.42 to −0.29 | 9 | −0.82 | −4.41 to 2.77 | 0.62 |
| | 2.3 Self-monitoring of behaviour. | −0.91 | −2.13 to 0.31 | 3 | −0.89 | −1.55 to −0.25 | 9 | −0.04 | −1.27 to 1.10 | 0.95 |
| | 4.1 Instruction on how to perform the behaviour. | −0.81 | −1.57 to −0.05 | 5 | −0.99 | −1.77 to −0.22 | 7 | 0.17 | −1.08 to 1.42 | 0.77 |
| | 9.1 Credible source. | −0.95 | −1.52 to −0.38 | 8 | −0.27 | −1.89 to 1.35 | 4 | −0.65 | −2.66 to 1.36 | 0.49 |
| | 9.2 Pros and cons. | −0.91 | −3.91 to 2.09 | 3 | −0.88 | −1.48 to −0.28 | 9 | −0.01 | −3.51 to 3.49 | 0.99 |

β, meta-regression coefficient; BCT, behaviour change technique; BMI, body mass index; MD, mean difference; N, number of trials.

factors observed may be unlikely to reflect changes occurring over longer periods. This review found reductions in blood pressure and total cholesterol following intervention, but in some instances, this might be mediated by pharmacological treatment. There are clear benefits of drug treatments in lowering blood pressure and cholesterol in primary prevention populations.[61 62]

Although this review focused on interventions for the primary prevention population, we also included trials that recruited a small minority of participants with some evidence of CVD. Including these trials might have biased the results, as health promotion interventions might have more positive effects in people with established CVD.[63–65]

In order to account for heterogeneity, we focused on trial level covariates and identified characteristics that might be associated with more favourable outcomes. When coding BCTs, we were limited by the lack of detail provided in reports. We only coded what was explicitly referred to in intervention descriptions and could be fitted to BCT taxonomy definitions.

This review suggested no association between the number of intervention sessions or intervention duration and improved outcomes. Quantity of sessions would not necessarily have a beneficial impact on outcomes unless additional sessions deliver BCTs that effectively influence behaviours. Few reports provided sufficient information to permit calculating duration for analysis. Increasing use of the Template for Intervention Description and Replication (TIDieR) checklist,[66] requiring intervention reports to detail the number and duration of sessions offered to participants, will be helpful for future reviews.

Our analyses suggested that using certain BCTs has a moderator effect on intervention outcomes. In terms of biomarkers of CVD risk, no BCTs were identified as being particularly likely to influence cholesterol levels, while including review of behaviour goals appeared to be associated with slightly worse blood pressure outcomes.

'Action planning' was associated with greater weight loss, while 'Instruction on how to perform the behaviour' was not. Both of these findings differ to those of a previous review,[17] perhaps because it focused only on interventions for obese individuals. The previous review also identified the BCTs of self-monitoring, relapse prevention/problem solving and prompt practice as beneficial to weight loss, but too few of the interventions included in the present review incorporated these BCTs for it to be possible to test their influence. A review of interventions promoting healthy eating and exercise also found that including the BCT of self-monitoring was associated with bigger changes in these behaviours.[18] Therefore, one explanation for the relatively limited effectiveness of the interventions reviewed in the present review is that they failed to include BCTs that were more likely to lead to health-promoting changes. A second possibility is that not all BCTs were delivered as the intervention designers intended. This cannot be ruled out as monitoring of treatment fidelity was rarely described in the included studies.

This review showed no association between the use of psychological theory and improved intervention outcomes. However, only a limited range of theories were employed—mostly the Transtheoretical Model and Social Cognitive Theory. A previous review also found that interventions based on these theories were not significantly more effective than interventions not explicitly based on theory.[13] A second issue is that the links between the psychological determinants specified by a theory and the BCTs employed in interventions were sometimes poorly articulated, with little evidence cited to justify choice of BCTs to change specific constructs. Furthermore, it was not always clear which BCTs were being used to target which behaviours as part of the MHBC interventions. Both this and previous reviews[13 67] found that reported theory use in intervention design was not as extensive as it could be. It is possible that interventions based on other theories or that more explicitly link theoretical constructs to select BCTs might be more effective.

Future trials need to test interventions that provide explicit links between intervention components (ie, theoretical basis, BCTs and intended mechanisms of action, intervention duration) and intervention outcomes as it is essential step towards understanding MHBC intervention effects. Higher priority should also be given to different population-level approaches to facilitate behaviour change.

## Limitations

The results of this review must be viewed with caution because of several limitations. First, the observed effects were heterogeneous; therefore, pooled estimates might be questionable. DerSimonian and Laird (DL)[22] random effects models were used. The DL method may lead to under-estimation of between-trial variance leading to narrower CIs in the presence of heterogeneity.[68 69] However, Thorlund *et al*[70] concluded that inferences concerning pooled effects were only infrequently influenced by the choice of between-trial variance estimator. The majority of trials included were undertaken in Europe (71%) and the United States (13%). Declines in CVD mortality and CVD risk have been observed in these countries, and the results should be considered in the context of these trends. Groups of BCTs may have synergistic effects on behaviour.[16] However, due to the relatively small numbers of studies and under-description of the BCTs used in interventions, it was not possible to explore the impact of clusters of BCTs on CVD risk factors, as too few studies used the same clusters of BCTs and measured the same outcome. Furthermore, the differences between subgroups and covariates (ie, theory use and BCTs) and effect size are observational and do not imply causality. Behavioural risk factors were assessed by self-report and so values were subject to social desirability and recall biases. Finally, as this review involved testing for the impact of MHBC interventions and intervention characteristics on intervention outcomes, we are aware of the need to adjust p values based on the number of

tests being made.[71] Although adjusting p values reduces type 1 error, it increases the chances of false negatives.[72] Furthermore, tests were examining independent hypotheses; therefore, p values were not adjusted.[73]

## CONCLUSION

Existing MHBC interventions delivered to individual participants in primary care appear to have limited effectiveness at reducing CVD risk and CVD risk factors over 12 months or longer. Trial reports need to provide explicit explanation of the intervention theory, content and delivery, including fidelity and care provided to the control group in order to understand why an intervention may or may not prove effective. This is essential for future development and evaluation of effective CVD prevention interventions.

**Correction notice** This paper has been amended since it was published Online First. Owing to a scripting error, some of the publisher names in the references were replaced with 'BMJ Publishing Group'. This only affected the full text version, not the PDF. We have since corrected theseerrors and the correct publishers have been inserted into the references.

**Acknowledgements** Open access for this article was funded by King's College London.

**Contributors** SA, AJW and MCG conceptualised and designed the study. SA and MCG performed the paper search. SA and LM performed the coding. SA wrote the first draft and all authors have read and made improvements of the contents and the wordings.

**Funding** MCG was supported by the National Institute for Health Research Biomedical Research Centre at Guy's and St Thomas' NHS Foundation Trust and King's College London. SA was supported by the Government of Saudi Arabia.

**Competing interests** None declared.

**Provenance and peer review** Not commissioned; externally peer reviewed.

**Data sharing statement** No additional data are available.

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
