## [Reviewer comments · BMJ Open]

ARTICLE DETAILS

TITLE (PROVISIONAL)	Multiple health behaviour change interventions for primary prevention of cardiovascular disease in primary care: systematic review and meta-analysis
AUTHORS	Alageel, Samah; Gulliford, Martin; McDermott, Lisa; Wright, Alison

VERSION 1 - REVIEW

REVIEWER	Dr Sam Merriel Centre for Academic Primary Care University of Bristol United Kingdom
REVIEW RETURNED	21-Dec-2016

GENERAL COMMENTS	The authors have presented a very thorough and methodologically sound systematic review and meta analysis on the effect of multiple health behaviour change interventions on reducing CVD risk in primary care. The addition of an analysis of the various underlying theories of behaviour change is important, and gives the reader a more in-depth understanding of the trials assessed in this review. The findings are consistent with similar systematic reviews in this field.
---

REVIEWER	George C Roush UCONN School of Medicine USA
REVIEW RETURNED	12-Jan-2017

GENERAL COMMENTS	*First of all, I apologize for not reviewing your paper sooner. I found myself in the midst of a major deadline. *Thank you for an interesting and important paper. *Why was control of diabetes omitted as one of the outcomes? It is an important part of the metabolic syndrome and your paper includes the other 3 components: hypertension, dyslipidemia, and obesity. At least this omission deserves an explanation. *The exact type of model is not described. Saying "random effects" is not sufficient. Was this DerSimonian-Laird? If so, this model gives too many false positives, particularly with smaller meta-analyses in the presence of heterogeneity (Cornell JE, Mulrow CD, Localio R, Stack CB, Meibohm AR, Guallar E, Goodman SN. Random-effects meta-analysis of inconsistent effects: a time for change. Ann Intern Med 2014; 160:267–270.) See attached paper. *Heterogeneity would be expected in this review. Unless I-squared is 0% (when tau will equal 0 and the p for heterogeneity will be non-significant, I-squared by itself is insufficient. Include tau and the P value for heterogeneity in all of the presentations in Table 3. If tau,
---

	the standard deviation of the main effect, is elevated relative to the main effect (e.g., tau/main effect > 1), this indicates heterogeneity and gives different information than I-squared. *In my opinion, a "leave one out" sensitivity analysis is necessary, particularly when there are fewer studies as in this case. Also, it may help to figure out the sources of heterogeneity by determining whether a particular study accounts for some or most of the heterogeneity. You might find that leaving out a particular study will yield a significant result. Hopefully, your software will operationalize this so it can be done conveniently. CMA software does allow this but probably others do as well. *Heterogeneity in some instances should be included as a limitation in the sections on "strengths and limitations" and should be mentioned in the abstract to alert the reader to this finding. *When examining heterogeneity for blood pressure, you could use meta-regression with Medication/no medication as a potential effect modifier. It gives you a formal test. *Meta-regression (all of the results in Table 4) is an observational type of analysis. Identify this as a limitation in the limitations sections. *On the other hand, a strength of your paper is that your main focus is on head-to-head analyses of randomized trials Mention this in the "strengths & limitations" section and probably in the discussion. *In general, in your paper, publication bias is probably not decisive when interpreting the results and drawing conclusions. However, when you are finding significant differences, it would be optimal to use multiple methods for evaluating this and not just Eggers because any one test may have limited power. A more comprehensive approach would be Eggers, Begg & Mazumdar, trim & fill, limiting analysis to larger or more precise studies, and/or cumulative meta-analysis (Leimu R, Koricheva J. Cumulative meta-analysis: a new tool for detection of temporal trends and publication bias in ecology. Proc Biol Sci 2004; 271:1961–1966.) Hopefully your software will operationalize this. CMA does have this feature as may others. (I've no financial or professional interest in CMA... just letting you know.) *Shouldn't "STUDY CHARACTERISTICS" page 21, line 44 thru page 22 line 9 go near the beginning of the results? *Discussion, 2nd paragraph, line 26,27 has a misspelled/unintelligible word. *Page 10, line 5, I think "and" is missing between "duration, types". The reviewer also provided a file in addition to these comments. Please contact the publisher for full details.
--	--

REVIEWER	Ligthart, Suzanne Academic Medical Center (AMC), Amsterdam, The Netherlands
REVIEW RETURNED	06-Feb-2017

GENERAL COMMENTS	General comments: This is an interesting, informative and very extensive review about the value of multiple health behavior change (MHBC) interventions for lowering cardiovascular risk factors in primary care. I have made some suggestions and comments below, that will hopefully add some further clarity of the paper. This study adds information to knowledge of a review of
--

2011/Ebrahim et al(5), in which only event and mortality numbers were considered. It looks at risk factors instead of events, which seems justified in the light of the baseline risk of the patients considered (relatively young, without CVD) and the duration of trials.

Title: In this manuscript, primary prevention of CVD is not studied, but risk factors for CVD. And: are all studies in primary care or also in unselected/community based populations?

Abstract: Throughout the abstract, 'primary prevention' is not mentioned, while 'primary care' is. In my view (and country), primary as well as secondary prevention is provided in primary care/by GP's and practice nurses. It is not clear from the abstract that primary prevention is studied.

P2 Line13: CVD-risk or CVD risk factors? In the objective/methods (P6/8) CVD-risk(-score) AND CVD risk factors are mentioned as primary outcomes but in the abstract only CVD risk factors.

Conclusion L 44: CVD-risk is not mentioned in the results-section. In primary care= for primary prevention?

Introduction: P4L50: primary prevention instead of primary care?

Methods: P7 L14: was there an age limit? In the flow diagram 'sample too young' is stated as an exclusion criterion.

P8 L 41: why 10% and where there any differences between the reviewers?

P9 L6-22: The how and why of the 'application of theory' and why these three measures of the TCS were used is not entirely clear to me.

Discussion: Generally: clear.

P28 L26: I think you are correctly aware (and: testes ->tests:). The argument that 'however tests were examining independent hypotheses, therefore the p-values were not adjusted' is incorrect. Rolling a dice gives a change of 1 out of 6 that 6 will come up. However, if 30 dices are thrown, totally independent of each other, the chance that at least one of them shows a 6 is very high...

Conclusion: maybe add 'care provided in the control group' as another factor that needs to be reported explicitly. Further: do the authors advice that more trials be done? And how? With regard to the results presented here. I would like to read some more about this in the discussion-section.

Tables/figures:

Table 1: Why name UK and US, while Sweden (5) and the Netherlands (4) also provide a substantial amount of studies? I would name all countries with 3 or more studies.

Table 4: these are a lot of (P-)values, maybe consider this to report in the supplementary files? It does not offer me very valuable information.

Supplementary table 1: I would personally like to see this table in the manuscript itself.

REVIEWER	RM Van den Berg-Vos Department of Neurology, OLVG West hospital, Jan Tooropstraat 164, 1061 AE Amsterdam, the Netherlands
REVIEW RETURNED	07-Feb-2017

GENERAL COMMENTS	Multiple health behaviour change interventions for primary prevention of cardiovascular disease in primary care: systematic review and meta-analysis The authors have performed a solid systematic review and meta-analysis of life style interventions for primary prevention of cardiovascular disease. I find the study design well described and the manuscript is well readable. The description of the data analysis is clear. I consider as strengths of the study:  • the inclusion criterion of > 12 months follow-up • the description of and study on the disentanglement of several characteristics of the behaviour change interventions and the application of the Theory Coding Scheme I consider as limitations the following points:  • the fact that the literature search was done until May 2015, which is already almost 2 years ago • I do not understand why the primary endpoints cardiovascular disease mortality and cardiovascular clinical events are not incorporated. The described literature concerning these endpoints (references 5,6 and 8) dates from 2011, 2008 and 2012 respectively. • Changes in CVD risk factors (page 19) and Discussion section (page 25): the authors report a modest but significant effect on diastolic RR and total cholesterol level but label these findings as not significant, which I think is contradictory, as they end with the conclusion "multiple health behaviour change interventions evaluated to date for the primary prevention of CVD may generally have very limited effects in reducing CVD-risk and CVD risk factors in primary care". On page 19 is stated that in 6 of the 12 studies on blood pressure patients used antihypertensive drugs (4 studies) and unspecified medication (2 studies) and in studies on cholesterol levels patients took lipid-lowering medication in 3 out of 10 studies. Isn't it better to exclude these studies in the meta-analysis to investigate whether multiple health behaviour change interventions apart from medication have an effect on cardiovascular risk factors in primary care? Further comments:  • Page 2 Abstract after line 46: I suggest to add a sentence to the Conclusion concerning the need to provide explanation of the intervention theory, content and delivery to understand why an intervention may or may not prove effective. • Page 5 line 9: please adjust the verb "inform" • Page 5 line 11 until 15: adjust the sentence "interventions .. until effective". It is not good readable • Page 5 line 43: what do the authors mean by a control theory, please give an explanation • Page 10 line 1: I presume that the word analyses is missing after the word meta-regression? • Page 17: concerning the paragraph about Risk of bias in included studies: I suggest that the authors describe in the text how much (high, moderate and low) bias the authors have ascertained to the included studies. I am not convinced that in this paragraph in the text
---

	the intention to treat analysis is the only important point...  • Page 18 table 3: What is meant with the annotation “none” in the row pooled effect size? Has it something to do with no specification or report of used medication, as I can read in the text? Please add this explanation tot the table • Page 23 line 22/23: replace than with compare with. Is the description of an increase in systolic blood pressure and less change in diastolic blood pressure correct? Isn't it more reasonable to assume a decrease and more change in the intervention arms? • Page 25 line 27/28: incorrect spelling of wlgithly?? ; do the authors mean “slightly”? • Page 25 line 41: remove “a” before word longer • Page 26 line 30: fewer reports??? than what? • Page 27 line 31: please avoid abbreviations TTM and SCT in the text here, have only been mentioned before once and I presume the reader does not remember... • Page 28 line 28: testes?? Do the authors mean tests? • Page 28 line 42: need instead of needs
--	---

VERSION 1 – AUTHOR RESPONSE

Reviewer: 1

Reviewer Name: Dr Sam Merriel

Institution and Country: Centre for Academic Primary Care, University of Bristol, United Kingdom

Please state any competing interests: None declared

Please leave your comments for the authors below

The authors have presented a very thorough and methodologically sound systematic review and meta-analysis on the effect of multiple health behaviour change interventions on reducing CVD risk in primary care. The addition of an analysis of the various underlying theories of behaviour change is important, and gives the reader a more in-depth understanding of the trials assessed in this review. The findings are consistent with similar systematic reviews in this field.

Thank you for this feedback.

Reviewer: 2

Reviewer Name: George C Roush

Institution and Country: UCONN School of Medicine, USA Please state any competing interests:

None

Please leave your comments for the authors below

*First of all, I apologize for not reviewing your paper sooner. I found myself in the midst of a major deadline.

*Thank you for an interesting and important paper.

*Why was control of diabetes omitted as one of the outcomes? It is an important part of the metabolic syndrome and your paper includes the other 3 components: hypertension, dyslipidemia, and obesity. At least this omission deserves an explanation.

Thank you for this comment. We have excluded trials of diabetes patient populations, therefore diabetes control outcomes were not included (page 8 and 9).

*The exact type of model is not described. Saying "random effects" is not sufficient. Was this

DerSimonian-Laird? If so, this model gives too many false positives, particularly with smaller meta-analyses in the presence of heterogeneity (Cornell JE, Mulrow CD, Localio R, Stack CB, Meibohm AR, Guallar E, Goodman SN. Random-effects meta-analysis of inconsistent effects: a time for change. *Ann Intern Med* 2014; 160:267–270.) See attached paper.

Thank you for this comment. The model is now specified in page 11 of the methods section. Now we add this: "DerSimonian and Laird (DL) random effects models were used. The DL method may lead to under-estimation of between trial variance leading to narrower confidence intervals in the presence of heterogeneity. However, Thorlund et al. Concluded that inferences concerning pooled effects were only infrequently influenced by the choice of between-trial variance estimator" to the limitation section page 32.

*Heterogeneity would be expected in this review. Unless I-squared is 0% (when tau will equal 0 and the p for heterogeneity will be non-significant, I-squared by itself is insufficient. Include tau and the P value for heterogeneity in all of the presentations in Table 3. If tau, the standard deviation of the main effect, is elevated relative to the main effect (e.g., tau/main effect > 1), this indicates heterogeneity and gives different information than I-squared.

Thank you, values for Tau² and the P value for heterogeneity have now been added to table 3.

*In my opinion, a "leave one out" sensitivity analysis is necessary, particularly when there are fewer studies as in this case. Also, it may help to figure out the sources of heterogeneity by determining whether a particular study accounts for some or most of the heterogeneity. You might find that leaving out a particular study will yield a significant result. Hopefully, your software will operationalize this so it can be done conveniently. CMA software does allow this but probably others do as well.

Thank you for this comment, we have conducted a leave-one-out sensitivity analysis for outcomes with significant heterogeneity (page 23-24).

*Heterogeneity in some instances should be included as a limitation in the sections on "strengths and limitations" and should be mentioned in the abstract to alert the reader to this finding.

Thank you for this comment, this has been explained in the limitations (page 32) and abstract sections (page2).

*When examining heterogeneity for blood pressure, you could use meta-regression with Medication/no medication as a potential effect modifier. It gives you a formal test.

Thank you, meta-regression analyses were used to examine the impact of medication use on blood pressure and serum total cholesterol outcomes, and the results are now shown on page 21.

*Meta-regression (all of the results in Table 4) is an observational type of analysis. Identify this as a limitation in the limitations sections.

Thank you for this comment. We now add, "the differences between subgroups and covariates (i.e. theory use and BCTs) and effect size are observational and do not imply causality." in the limitations section page 32.

*On the other hand, a strength of your paper is that your main focus is on head-to-head analyses of randomized trials. Mention this in the "strengths & limitations" section and probably in the discussion.

Thank you for your comment. We now added this in the strengths and limitations section (page 32).

*In general, in your paper, publication bias is probably not decisive when interpreting the results and drawing conclusions. However, when you are finding significant differences, it would be optimal to use multiple methods for evaluating this and not just Eggers because any one test may have limited power. A more comprehensive approach would be Eggers, Begg & Mazumdar, trim & fill, limiting analysis to larger or more precise studies, and/or cumulative meta-analysis (Leimu R, Koricheva J. Cumulative meta-analysis: a new tool for detection of temporal trends and publication bias in ecology. Proc Biol Sci 2004; 271:1961–1966.) Hopefully your software will operationalize this. CMA does have this feature as may others. (I've no financial or professional interest in CMA... just letting you know.)

Thank you for your comment. When bias existed, we used the 'trim and fill' method to adjust for publication bias. The data were unchanged as the results showed no trimming was performed. We now added, "if bias existed, the trim and fill method was used to adjust for publication bias" to the methods section (page 12), and "The results of "trim and fill" method indicated that the weighted mean did not change despite the existence of publication bias (Egger's test $P=0.002$)." in the results section (page 22).

*Shouldn't "STUDY CHARACTERISTICS" page 21, line 44 thru page 22 line 9 go near the beginning of the results?

Thank you, this section presents the effect of intervention components on effectiveness. We now refer to "intervention components".

*Discussion, 2nd paragraph, line 26,27 has a misspelled/unintelligible word.

Thank you, this has been changed.

*Page 10, line 5, I think "and" is missing between "duration, types".

Thank you, this has been changed.

Reviewer: 3

Reviewer Name: S. Ligthart

Institution and Country: Academic Medical Center (AMC), Amsterdam, The Netherlands Please state any competing interests: None declared

Please leave your comments for the authors below

General comments:

This is an interesting, informative and very extensive review about the value of multiple health behavior change (MHBC) interventions for lowering cardiovascular risk factors in primary care. I have made some suggestions and comments below, that will hopefully add some further clarity of the paper.

This study adds information to knowledge of a review of 2011/Ebrahim et al(5), in which only event and mortality numbers were considered. It looks at risk factors instead of events, which seems justified in the light of the baseline risk of the patients considered (relatively young, without CVD) and the duration of trials.

Thank you for this feedback.

Title: In this manuscript, primary prevention of CVD is not studied, but risk factors for CVD. And: are all studies in primary care or also in unselected/community based populations?

Thank you for this comment. We have included trials of interventions for CVD-free populations aimed at reducing CVD-risk. We only included interventions in primary care, trials of interventions conducted in other settings were excluded (page 8).

Abstract: Throughout the abstract, 'primary prevention' is not mentioned, while 'primary care' is. In my view (and country), primary as well as secondary prevention is provided in primary care/by GP's and practice nurses. It is not clear from the abstract that primary prevention is studied.

Thank you for this comment. We have now updated the abstract (line 2 and line 7).

P2 Line13: CVD-risk or CVD risk factors? In the objective/methods (P6/8) CVD-risk (-score) AND CVD risk factors are mentioned as primary outcomes but in the abstract only CVD risk factors.

Thank you, we did not include the estimate for CVD risk scores in the abstract because this was based on only two studies.

Conclusion L 44: CVD-risk is not mentioned in the results-section. In primary care= for primary prevention?

Thank you, we now say: "MHBC interventions delivered to CVD-free participants in primary care did not appear to have quantitatively important effects on CVD risk factors."

Introduction: P4L50: primary prevention instead of primary care?

Thank you, this has been changed.

Methods: P7 L14: was there an age limit? In the flow diagram 'sample too young' is stated as an exclusion criterion.

Thank you, we excluded trials of participants less than 18 years old. We added, "Trials that recruited an adult population (>18 years old) free of CVD were included." to the participants section and a change was made to the flow diagram.

P8 L 41: why 10% and where there any differences between the reviewers?

Thank you, this now added to the search strategy section in page 9.

P9 L6-22: The how and why of the 'application of theory' and why these three measures of the TCS were used is not entirely clear to me.

Thank you for this comment. We employed similar approaches to operationalising different levels of the use of theory in intervention design as in a previous systematic review (Prestwich, Sniehotta, Whittington et al, 2014). We now add on page 10-11, "We used three measures to capture the extent of theory use, as employed in a previous review. The first concerned whether the intervention was explicitly based on a theory or combination of theories or predictors (TCS item 5). Secondly, we assessed the degree to which each BCT reported as part of the intervention was linked to a theory-

relevant construct (scored +2 for the ideal scenario of “yes” to TCS item 7 (all intervention techniques explicitly linked to at least one theory-relevant construct), +1 for studies coded “yes” for TCS item 8 (at least one, but not all, intervention techniques explicitly linked to at least one theory-relevant construct) and/or TCS item 9 (group of BCTs are linked to a group of constructs) and 0 for studies coded “no” for all of items 7-9. Finally, we rated ; and the extent to which all constructs explicitly targeted by BCTs. This was scored +2 for the ideal scenario of “yes” to TCS item 10 (all theory-relevant constructs explicitly linked to at least one BCT), +1 for “yes” to TCS item 9 (group of BCTs are linked to a group of constructs) and/or item 11 (at least one, but not all, theory relevant constructs are explicitly linked to at least one BCT) and 0 for interventions coded “no” to all of items 9-11.”

Discussion: Generally: clear.

P28 L26: I think you are correctly aware (and: testes ->tests:). The argument that ‘however tests were examining independent hypotheses, therefor the p-values were not adjusted’ is incorrect. Rolling a dice gives a change of 1 out of 6 that 6 will come up. However, if 30 dices are thrown, totally independent of each other, the chance that at least one of them shows a 6 is very high...

Thank you for this comment. We now add this: “Although adjusting p-values reduces type 1 error, it increases the chances of false negatives. Furthermore, tests were examining independent hypotheses, therefore the p-values were not adjusted.” to the limitations section.

Conclusion: maybe add ‘care provided in the control group’ as another factor that needs to be reported explicitly.

Thank you, this has now been added to the conclusion (page 33).

Further: do the authors advice that more trials be done? And how? With regard to the results presented here. I would like to read some more about this in the discussion-section.

Thank you, we now added: “Future trials need to test interventions that provide explicit links between intervention components (i.e. theoretical basis, BCTs and intended mechanisms of action, intervention duration) and intervention outcomes as it is essential step towards understanding MHBC intervention effects. Higher priority should also be given to different population-level approaches to facilitate behaviour change” to the discussion section (page 31).

Tables/figures:

Table 1: Why name UK and US, while Sweden (5) and the Netherlands (4) also provide a substantial amount of studies? I would name all countries with 3 or more studies.

Thank you for this comment, Sweden and the Netherlands have been added to table 1.

Table 4: these are a lot of (P-) values, maybe consider this to report in the supplementary files? It does not offer me very valuable information.

Thank you for this comment. We considered table 4 to be an essential part of the review, we could remove the P values column if that would improve the table’s readability.

Supplementary table 1: I would personally like to see this table in the manuscript itself.

Thank you for this comment. As we are limited with five figures and tables, we decided to present this table as a supplementary table.

Reviewer: 4

Reviewer Name: RM Van den Berg-Vos

Institution and Country: Department of Neurology, OLVG West hospital, Jan Tooropstraat 164, 1061 AE Amsterdam, the Netherlands Please state any competing interests: None declared

Please leave your comments for the authors below

Multiple health behaviour change interventions for primary prevention of cardiovascular disease in primary care: systematic review and meta-analysis

The authors have performed a solid systematic review and meta-analysis of life style interventions for primary prevention of cardiovascular disease. I find the study design well described and the manuscript is well readable. The description of the data analysis is clear.

Thank you for your feedback.

I consider as strengths of the study:

- The inclusion criterion of > 12 months follow-up

Thank you, this have been added to the strengths and limitation section.

- The description of and study on the disentanglement of several characteristics of the behaviour change interventions and the application of the Theory Coding Scheme

Thank you, this have been added to the strengths and limitation section.

I consider as limitations the following points:

- The fact that the literature search was done until May 2015, which is already almost 2 years ago

Thank you. We have now updated the search until February 2017, as shown on page 9 and figure 1 (page 14). This resulted in the addition of four trials to the review.

- I do not understand why the primary endpoints cardiovascular disease mortality and cardiovascular clinical events are not incorporated. The described literature concerning these endpoints (references 5,6 and 8) dates from 2011, 2008 and 2012 respectively.

Thank you for this comment, only one study in 2015 reported clinical outcomes, therefore long term outcomes were not examined in this review (page 9).

- Changes in CVD risk factors (page 19) and Discussion section (page 25): the authors report a modest but significant effect on diastolic RR and total cholesterol level but label these findings as not significant, which I think is contradictory, as they end with the conclusion "multiple health behaviour change interventions evaluated to date for the primary prevention of CVD may generally have very limited effects in reducing CVD-risk and CVD risk factors in primary care".

Thank you for the comment, we now add: "Although pooled effects of interventions on risk factors were statistically significant but clinically modest." to the discussion section on page 29.

- On page 19 is stated that in 6 of the 12 studies on blood pressure patients used antihypertensive drugs (4 studies) and unspecified medication (2 studies) and in studies on cholesterol levels patients took lipid-lowering medication in 3 out of 10 studies. Isn't it better to exclude these studies in the meta-analysis to investigate whether multiple health behaviour change interventions apart from medication have an effect on cardiovascular risk factors in primary care?

Thank you, meta-regression was used to examine the impact of medication use on blood pressure and serum total cholesterol outcomes and the results are shown in page 20.

Further comments:

- Page 2 Abstract after line 46: I suggest to add a sentence to the Conclusion concerning the need to provide explanation of the intervention theory, content and delivery to understand why an intervention may or may not prove effective.

Thank you for this comment. We now add, "Better reporting of interventions' rationale, content and delivery is essential to understanding their effectiveness." to the conclusion section in the abstract (page 2).

- Page 5 line 9: please adjust the verb "inform"

Thank you. This has been changed.

- Page 5 line 11 until 15: adjust the sentence "interventions .. until effective". It is not good readable

Thank you, we now say: "Interventions are likely to be more effective when they systematically target psychological determinants of behaviour" on page 6.

- Page 5 line 43: what do the authors mean by a control theory, please give an explanation

Thank you, we now say: "while interventions designed to modify physical activity and/or diet were more effective when they included self-monitoring plus one of the four following behaviour change techniques: prompting intention formation, specific goal setting, review of behavioural goals or providing feedback on performance".

- Page 10 line 1: I presume that the word analyses is missing after the word meta-regression?

Thank you. This has now been changed.

- Page 17: concerning the paragraph about Risk of bias in included studies: I suggest that the authors describe in the text how much (high, moderate and low) bias the authors have ascertained to the included studies. I am not convinced that in this paragraph in the text the intention to treat analysis is the only important point...

Thank you for this comment. The Cochrane risk of bias tool does not give an overall score of bias for each study. We now added more details to explain the risk of bias of the included studies (page 19) and the assessment results in supplementary table 3.

- Page 18 table 3: What is meant with the annotation "none" in the row pooled effect size? Has it something to do with no specification or report of used medication, as I can read in the text? Please add this explanation tot the table

Thank you, we have now added an explanation to table 3.

- Page 23 line 22/23: replace than with compare with. Is the description of an increase in systolic blood pressure and less change in diastolic blood pressure correct? Isn't it more reasonable to assume a decrease and more change in the intervention arms?

Thank you, the use of certain BCTs was associated with worse blood pressure outcomes, as explained in the discussion section page 30.

- Page 25 line 27/28: incorrect spelling of wlgithly?? ; do the authors mean “slightly”?

Thank you. This has now been changed.

- Page 25 line 41: remove “a” before word longer

Thank you. This has now been changed.

- Page 26 line 30: fewer reports??? than what?

Thank you. This has now been changed.

- Page 27 line 31: please avoid abbreviations TTM and SCT in the text here, have only been mentioned before once and I presume the reader does not remember...

Thank you. This has now been changed.

- Page 28 line 28: testes?? Do the authors mean tests?

Thank you. This has now been changed.

- Page 28 line 42: need instead of needs

Thank you. This has now been changed.

VERSION 2 – REVIEW

REVIEWER	Ligthart, Suzanne AMC/Academic Medical Center, Amsterdam, the Netherlands
REVIEW RETURNED	21-Mar-2017

GENERAL COMMENTS	I think the manuscript has improved substantially, and I am impressed by the amount of work that has been done by the authors. The result is an extensive and thorough review and meta-analysis, that will be usefull to the readers. I still think that the problem of multiple testing exists and would interpret the findings of $p=0.04$ (twice) very cautiously; this has been addressed now in the discussion.
---